# LABAs and p38MAPK Inhibitors Reverse the Corticosteroid-Insensitivity of IL-8 in Airway Smooth Muscle Cells of COPD

**DOI:** 10.3390/jcm8122058

**Published:** 2019-11-22

**Authors:** Jürgen Knobloch, David Jungck, Juliane Kronsbein, Erich Stoelben, Kazuhiro Ito, Andrea Koch

**Affiliations:** 1Medical Clinic III for Pneumology, Allergology and Sleep Medicine, Bergmannsheil University Hospital, Ruhr-University Bochum, Bürkle-de-la-Camp-Platz 1, 44789 Bochum, Germany; David.Jungck@BethelNet.de (D.J.); juliane.kronsbein@bergmannsheil.de (J.K.); 2Department of Internal Medicine II, Pneumology, Allergology and Respiratory Medicine, Bethel Teaching Hospital, 12207 Berlin, Germany; 3Lungclinic, Thoracic Surgery, Hospital of Cologne, Private University Witten Herdecke, 51109 Cologne, Germany; StoelbenE@kliniken-koeln.de; 4National Heart and Lung Institute, Imperial College London, London SW3 6LY, UK; k.ito@imperial.ac.uk; 5Zürcher RehaZentren Davos, 7272 Davos-Clavadel, Switzerland; Ludwig-Maximilians-University of Munich (LMU) and DZL (German Center of Lung Science), 81377 Munich, Germany; andrea.koch@zhreha.ch

**Keywords:** COPD phenotypes, reversion of corticosteroid resistance, p38MAPK isoforms, long-acting-β2-agonist (LABA), non-type 2 inflammation

## Abstract

Airway inflammation in chronic obstructive pulmonary disease (COPD) is partially insensitive/resistant to inhaled corticosteroids (ICS). ICS plus bronchodilator therapy has been discussed for COPD phenotypes with frequent exacerbations and participation of corticosteroid-sensitive type 2/eosinophilic inflammation. Neutralization of non-type 2/IL-8-associated airway inflammation by reversion of its corticosteroid-resistance might be a future strategy for other phenotypes. Human airway smooth muscle cells (HASMCs) produce corticosteroid-insensitive IL-8 in response to TNFα or LPS in stable disease stages or bacteria-induced exacerbations, respectively. p38-mitogen-activated-protein-kinases (p38MAPKs) are alternative therapeutic targets. Hypothesis: long-acting-β2-agonists (LABAs) reverse the corticosteroid-insensitivity of IL-8 by p38MAPK inhibition in HASMCs. Cultivated HASMCs from COPD subjects were pre-incubated with formoterol, salmeterol, fluticasone-propionate, BIRB796 (p38MAPKα, -γ, -δ inhibitor), and/or SB203580 (p38MAPKα and -β inhibitor) before stimulation with TNFα or LPS. IL-8 and MAPK-activities were measured by ELISA. Formoterol, salmeterol, and fluticasone did not or hardly reduced TNFα- or LPS-induced IL-8. BIRB796 and SB203580 reduced TNFα-induced IL-8. SB203580 reduced LPS-induced IL-8. Fluticasone/formoterol, fluticasone/salmeterol, and fluticasone/BIRB796, but not fluticasone/SB203580 combinations, reduced TNFα-induced IL-8 stronger than single treatments. All combinations including fluticasone/SB203580 reduced LPS-induced IL-8 stronger than single treatments. TNFα induced p38MAPKα and -γ activity. LPS induced p38MAPKα activity. Formoterol reduced TNFα-induced p38MAPKγ and LPS-induced p38MAPKα activity. LABAs reverse the corticosteroid-insensitivity of IL-8 in airway smooth muscles via p38MAPKγ in stable disease and via p38MAPKα in exacerbations. Our pre-clinical data indicate a utility for also adding ICS in non-type 2 inflammatory COPD phenotypes to bronchodilator therapy. Depending on phenotype and disease stage, isoform-specific p38MAPK blockers might also reverse corticosteroid-resistance in COPD.

## 1. Introduction

Airway inflammation is the hallmark of chronic obstructive pulmonary disease (COPD) pathophysiology and progression in stable and exacerbated disease stages because it induces airway remodeling processes that impair lung function. The nature of the remodeling processes depends on disease phenotypes but includes irreversible airway obstruction, an increase in airway smooth muscle mass, lung tissue fibrosis and/or destruction of the alveolar walls (emphysema). Anti-inflammatory therapies are desirable because they would prevent the induction of remodeling processes but are not yet available for most COPD phenotypes and disease stages. COPD therapy is complicated by the partial resistance/insensitivity of the airway inflammation to inhaled corticosteroids (ICS) [1]. ICS use increases the risk of pneumonia in COPD [2]. Therefore, a careful assessment of ICS use in COPD is required, e.g., by identifying responder phenotypes.

Current guidelines recommend a long-acting muscarinic antagonist (LAMA)/long-acting β_2_-agonists (LABA) bronchodilator combination for symptomatic therapy [3]. Compared with monotherapies, LABA/ICS combinations are more effective in slowing the rate of lung function decline and preventing exacerbations [4]. Therefore, an ICS might be added to the LAMA/LABA therapy in frequent exacerbator phenotypes. This is currently recommended for phenotypes with high blood eosinophil counts that provide evidence for a participation of corticosteroid-sensitive type 2 inflammation [4].

COPD airway inflammation is perpetuated by the production of corticosteroid-sensitive and -insensitive cytokines and chemokines in response to inflammatory stimuli like TNFα and pathogen-associated molecular patterns (PAMPs) like lipopolysaccharide (LPS) from gram-negative bacteria [5,6,7,8]. The corticosteroid-insensitivity particularly applies to IL-8 (CXCL8), which is a central chemokine in non-type 2 (neutrophilic) airway inflammation that is present and critical in most likely all COPD phenotypes and disease stages and a target for therapeutic strategies [7,8,9,10]. The production of IL-8 and other COPD key cytokines are regulated by p38-mitogen-activated-protein kinase (p38MAPK) [7]. All four p38MAPK isoforms (α, β, γ, δ) are expressed in the lung tissue of COPD subjects and might contribute to non-type 2 inflammation [11].

Promising future causal therapy strategies to neutralize non-type 2 airway inflammation are (1) the drug-induced reversion of its corticosteroid-resistance to improve the effects of ICS, and (2) therapeutic inhibition of the activity of pro-inflammatory protein kinases, e.g., with narrow spectrum kinase inhibitors (NSKIs) that target a certain panel of pro-inflammatory protein kinases including p38MAPKα and -γ [7,12].

We hypothesized that LABAs reverse the corticosteroid-resistance in COPD through p38MAPK inhibition, thereby improving the effects of fluticasone on non-type 2 and IL-8-associated airway inflammation. If so, this would support a utility for ICS and, therefore, a causal therapy independent from type 2 inflammation for all COPD phenotypes either in combination with LABAs or with p38MAPK inhibitors.

Human airway smooth muscle cells (HASMCs) have been identified as an important source of COPD-related cytokines and chemokines that link corticosteroid-resistant non-type 2 airway inflammation to remodeling processes [6,13]. This includes endothelin-1 and GM-CSF, both of which are involved in the regulation of HASMC proliferation and, therefore, contribute to the increase in airway smooth muscle mass, leading to bronchial obstruction [13]. However, the most important cytokine might be IL-8 [6]. IL-8 recruits neutrophils into the lung tissue where they secrete several serine proteases, e.g., neutrophil elastase and metalloproteases, all of which are central triggers of alveolar destruction [1,5].

TNFα-induced IL-8 is completely insensitive to dexamethasone and fluticasone in HASMCs of COPD subjects but sensitive to these corticosteroids in HASMCs from subjects without COPD [7,8]. LPS-induced IL-8 is almost insensitive to fluticasone in HASMCs of COPD patients but sensitive to fluticasone in HASMCs from patients without COPD [7]. This suggests (1) important contributions of HASMCs to COPD non-type 2 airway inflammation and corticosteroid-resistance, and (2) suitability of HASMCs as therapeutic targets and as an experimental model to investigate our hypothesis.

We investigated whether LABAs and inhibitors of p38MAPK and related MAPK families (ERK, JNK) can reverse the corticosteroid-insensitivity in HASMCs of COPD. We induced the inflammatory IL-8 response in cultivated HASMCs with TNFα or LPS, representing culture models of non-type 2 airway inflammation in COPD [7]. TNFα is increased in the lungs of subjects with stable COPD, and HASMCs are permanently/repeatedly exposed to this pro-inflammatory factor. Therefore, the stimulation of cultivated HASMCs from COPD subjects with TNFα represents a culture model of stable disease stages [7]. In acute exacerbations caused by gram-negative bacteria, HASMCs get exposed to LPS. Therefore, the stimulation of cultivated HASMCs from COPD subjects with LPS represents a culture model of an exacerbated disease stage [7]. We investigated the effects of the LABAs formoterol and salmeterol and the corticosteroid fluticasone-propionate (all approved for COPD) and of the MAPK inhibitors BIRB796 (inhibitor of p38MAPKα, -γ, -δ isoforms), SB203580 (p38MAPKα and -β inhibitor), PD098059 (inhibitor of ERK activation), and SP600125 (inhibitor of JNK).

## 2. Material and Methods

The study population consisted of four current smokers and one ex-smoker (two female, three male) with COPD (GOLD stages I–III) undergoing lung resection for carcinoma (age (y, mean ± SEM): 70 ± 2.4; pack–years: 51.4 ± 9.2; FEV_1_ (% pred.): 63.9 ± 5.1; FEV_1_/FVC: 59.7 ± 3.1). For HASMC isolation, tumor-free bronchus tissue was used as previously described [6,7,8]. Subjects did not have histories of allergies and asthma, did not use corticosteroids, and did not report acute infections during the four weeks preceding the test. COPD was diagnosed and classified according to the NIH guidelines and the GOLD standard. The study was approved by the Ethics Committees of the Universities of Cologne (02-004) and Bochum (4257-12), Germany.

Isolation, characterization, and cultivation of HASMCs from COPD patients were performed as described [6,7,8]. Experiments were done with cells in culture passages two to five. Formoterol, salmeterol, fluticasone propionate, and BIRB796 (all RespiVert Ltd., London, UK), SB203580, PD098059, and SP600125 (all Sigma, Munich, Germany) were dissolved in DMSO. HASMCs in 6-well plates with about 90% confluence were serum-deprived for 24 h [6,7,8] and treated with the drugs alone or in combination 2 h before stimulation with LPS (Sigma, Munich, Germany), or TNFα (R&D Systems, Minneapolis, MN, USA).

IL-8 concentrations were measured in the supernatants by ELISA (R&D Systems, Minneapolis, DY208) after 72 h, as described [7,8]. The activities of p38MAPKα (T180 and Y182 phosphorylated p38MAPKα), p38MAPKγ (T183 and Y185 phosphorylated p38MAPKγ), and p38MAPKδ (T180 and Y182 phosphorylated p38MAPKδ) were measured in relation to the total expression levels of the respective isoforms by ELISA in cell lysates according to the instructions of the manufacturer (R&D Systems, Minneapolis, MN, USA; DYC8691B-2; DYC869-2; DYC1347-2; DYC1664-2; DYC1519-2; DYC2124-2). We did not investigate p38MAPKβ activity because a corresponding test for this isoform was not commercially available.

## 3. Results

Both TNFα and LPS induced IL-8 release from HASMCs of COPD subjects (Figure 1). Fluticasone did not significantly reduce TNFα- or LPS-induced IL-8 (Figure 1; Table 1), confirming that IL-8 is a corticosteroid-insensitive gene in HASMCs of COPD subjects. Salmeterol and formoterol, both, did not reduce TNFα- or LPS-induced IL-8 in single treatments (Figure 1; Table 1). The combination of fluticasone and salmeterol reduced both TNFα- and LPS-induced IL-8 and was, therefore, superior to the respective single treatments (Figure 1; Table 1). Accordingly, the combination of fluticasone and formoterol also reduced both TNFα- and LPS-induced IL-8 and was superior to the respective single treatments (Figure 1; Table 1). There were no statistically significant differences between the effects of formoterol and salmeterol (Figure 1). This demonstrates that the two LABAs reversed the corticosteroid-insensitivity of IL-8 in primary HASMC culture models of stable and exacerbated COPD.

In contrast to fluticasone, the p38MAPKα, -γ, -δ inhibitor BIRB796, the p38MAPKα and -β inhibitor SB203580, the inhibitor of ERK activation PD098059, and the JNK inhibitor SP600125 all reduced TNFα-induced IL-8 in a concentration-dependent manner (Figure 2A–D; Table 1). In combination with fluticasone, only BIRB796 but not the other MAPK inhibitors caused a significantly stronger IL-8 reduction compared to the respective single treatments (Figure 2A–D; Table 1). TNFα induced p38MAPKα and -γ activity (Figure 3). Phosphorylated (active) p38MAPKδ was not detectable. Formoterol reduced TNFα-induced p38MAPKγ but not p38MAPKα activity (Figure 3). These data indicate that all three MAPK families, p38MAPK, ERK, and JNK, are involved in the regulation of the production of IL-8 but that only the p38MAPKγ isoform is involved in the mechanism of the corticosteroid insensitivity of IL-8 in this cell culture model of stable COPD. They further indicate that formoterol reverses the corticosteroid-insensitivity of IL-8 by blocking p38MAPKγ activity in this experimental model.

SB203580, PD098059, and SP600125 but not BIRB796 reduced LPS-induced IL-8 in a concentration-dependent manner (Figure 2E–H; Table 1). In combination with fluticasone, SB203580 and BIRB796 but not the other MAPK inhibitors caused a significantly stronger IL-8 reduction compared to the respective single treatments (Figure 2E–H; Table 1). LPS induced p38MAPKα but not p38MAPKγ activity (Figure 3). Phosphorylated (active) p38MAPKδ was not detectable. Formoterol reduced LPS-induced p38MAPKα activity (Figure 3A). These data indicate that p38MAPKα, ERK, and JNK are involved in the regulation of the expression of IL-8 but that only p38MAPKα (but not the γ isoform) is involved in the mechanism of the corticosteroid insensitivity of IL-8 in this cell culture model of exacerbated COPD. They further indicate that formoterol reverses the corticosteroid-insensitivity of IL-8 by blocking p38MAPKα activity in this experimental model. However, due to technical limitations (see above), a role for p38MAPKβ cannot be excluded.

## 4. Discussion

Our data indicate that two LABAs, formoterol and salmeterol, both of which are frequently used in COPD therapy, can reverse the corticosteroid-insensitivity of IL-8 in COPD-relevant pre-clinical HASMC culture models. IL-8-associated non-type 2 (neutrophilic) airway inflammation is central in most likely all COPD disease stages and phenotypes. Thus, our pre-clinical data indicate for a utility of ICS in addition to standard LAMA/LABA therapy, also in phenotypes beyond those with type 2/eosinophilic airway inflammation, e.g., in patients with frequent exacerbations at LAMA/LABA. However, it has to be considered that information about other LABAs and u-LABAs and corticosteroids other than fluticasone that are approved for COPD in this context are lacking.

There are several clinical trials and meta-analyses that suggest a superiority of LABA/ICS combinations to respective monotherapies regarding exacerbation rates, lung function decline, and health status in COPD [2]. Our data might explain this superiority by an increased anti-inflammatory efficacy of the ICS through the reversion of the corticosteroid resistance by LABAs. The combined salmeterol/fluticasone therapy significantly reduced sputum IL-8 levels stronger compared with tiotropium monotherapy [14]. Although this study did not compare to salmeterol and fluticasone monotherapies, it demonstrated that a LABA/ICS combination could reduce IL-8 in COPD subjects, which matches our ex vivo data.

To our knowledge, our study is the first that has addressed the question of the effects of LABAs on the efficacy of corticosteroids in cells of the lung from COPD patients. This is important because local cells are the primary target of inhaled drugs. There are a couple of studies on systemic cells that, in part, support our data. In peripheral blood mononuclear cells (PBMCs), a fraction that contains various circulating immune cells, formoterol but not salmeterol, reversed the corticosteroid-insensitivity in cells from COPD subjects [15]. This indicates that the effects of different LABAs on the anti-inflammatory efficacy of corticosteroids depend on the cell type. Mechanistically, this effect of formoterol was explained by the inhibition of PI3-kinase-δ and subsequent re-activation of histone-deacytylase-2, a co-factor required for the activity of the glucocorticoid receptor, and by increased recruitment of the glucocorticoid receptor from the nucleus to the cytoplasm making it available for corticosteroids [15]. Formoterol can reverse the corticosteroid-insensitivity of IL-8 in PBMCs of severe asthma. As the underlying mechanism, it has been suggested that formoterol blocks p38MAPKγ activity, thereby preventing the pathological hyperphosphorylation of the glucocorticoid receptor that keeps it in the nucleus [16]. A following study also proposed a role for p38MAPKα in this context [17]. The endogenous p38MAPK antagonist MAPK-phosphatase-1 (MKP-1) could be a central player in the reversion of the corticosteroid insensitivity. Both formoterol and salmeterol have been shown to increase the fluticasone-induced upregulation of MKP-1 and the subsequent suppression of TNFα-induced IL-8 in HASMCs of undefined patients [18]. Reduced expression and/or activity of MKP-1 might contribute to the mechanism of the corticosteroid-insensitivity of IL-8 in HASMCs of COPD subjects and might be reversed by LABAs and bypassed by p38MAPK inhibitors.

These ex vivo data from severe asthma and HASMCs support our conclusions that the blockade of p38MAPK activity increases the sensitivity of IL-8 to fluticasone. They further support our findings regarding the central role of different p38MAPK isoforms in the corticosteroid-insensitivity of IL-8 in non-type 2 inflammation in COPD. Our data indicate that the reversion of the corticosteroid-insensitivity of IL-8 by formoterol depends on its inhibitory effects on different p38MAPK isoforms, p38MAPKα and -γ. We have provided evidence that TNFα, an inflammatory cytokine that is central for the perpetuation of non-type 2 airway inflammation in stable COPD, induces the activity of p38MAPKγ in HASMCs. We conclude that in stable disease stages, the corticosteroid resistance of IL-8 might be a direct consequence of p38MAPKγ activation by the inflammatory environment. In agreement, drug-induced inhibition of p38MAPKγ enabled fluticasone to reduce IL-8 in this model. In contrast, LPS, a gram-negative bacterial endotoxin relevant in exacerbated disease stages, induced p38MAPKα but not -γ activity, and blockade of p38MAPKα increased the effects of fluticasone on IL-8 in this model. We conclude that different p38MAPK isoforms are involved in the corticosteroid resistance of non-type 2/IL-8-associated inflammation in different disease stages. Notably, formoterol was able to block both isoforms and to increase the efficacy of fluticasone on IL-8 in both models. This indicates that formoterol can reverse corticosteroid insensitivity in stable and exacerbated disease.

Finally, our pre-clinical data suggest a utility for protein kinase inhibitors as causal therapy for corticosteroid-resistant airway inflammation in COPD. First, the individual inhibition of all MAPK families, p38MAPK, ERK, and JNK, reduced corticosteroid-insensitive IL-8 in both culture models. This might suggest protein kinase inhibitors as a causal therapeutic option for COPD that neutralizes non-type 2 airway inflammation without the need for ICS. However, the inhibition of single protein kinases or isoforms has proven to be ineffective in clinical settings because of redundancy in the functions of several kinases. On the other hand, targeting too many kinases simultaneously bears the risk of causing unwanted side effects because most kinases have various physiological functions [19]. Therefore, targeting a narrow spectrum of protein kinases might be a promising strategy, and our data suggest drugs that block—amongst others—p38MAPKα and -γ. Indeed, a drug that fulfills these criteria, RV568, showed a clinical benefit in COPD in a first small clinical trial [12]. A therapy strategy with narrow spectrum kinase inhibitors is currently in early clinical development ([12]; 25 September 2019: http://pulmatrix.com/pipeline.html).

Second, drug-induced inhibition of p38MAPK isoforms increase the efficacy of fluticasone on IL-8. This suggests that in combination with a corticosteroid, protein kinase inhibitors could have increased anti-inflammatory efficacy. If they target p38MAPK isoforms, they might not only directly block the production of cytokines in non-type 2 inflammation but also reverse their corticosteroid resistance and improve the efficacy of ICS. According to our data, research and development in this context might go in two directions: (1) targeting of specific p38MAPK isoforms might lead to the development of drug combinations for specific phenotypes and disease stages; (2) the simultaneous targeting of p38MAPKα and -γ, and possibly of other isoforms, might be suitable for a broad spectrum of COPD phenotypes. Although these strategies of reversing the corticosteroid resistance in COPD are yet far away from clinical development, our data strongly suggest clinical research activity in this direction.

HASMCs were isolated from lung resections of patients with COPD and lung cancer. Although IL-8 is corticosteroid-sensitive in HASMCs from lung cancer patients without COPD [7,8], and although the cells were prepared from tumor-free tissue, we cannot exclude the influence of lung cancer on our data. Even though lung cancer is a frequent co-morbidity of COPD, this is a limitation of the study.

## 5. Conclusions

Our pre-clinical data suggest (1) that LABAs can reverse the corticosteroid resistance of non-type 2 and IL-8-associated neutrophilic airway inflammation in COPD and, consequently, the further investigation of ICS utility in combination with bronchodilators also in COPD phenotypes beyond those with type 2/eosinophilic inflammation in clinical trials; and (2) a possible utility of protein kinase inhibitors that target p38MAPKα and/or -γ in combination with ICS as a strategy to neutralize non-type 2 corticosteroid resistant airway inflammation in a broad spectrum or in specific phenotypes and disease stages of COPD, respectively.

## Figures and Tables

**Figure 1 jcm-08-02058-f001:**
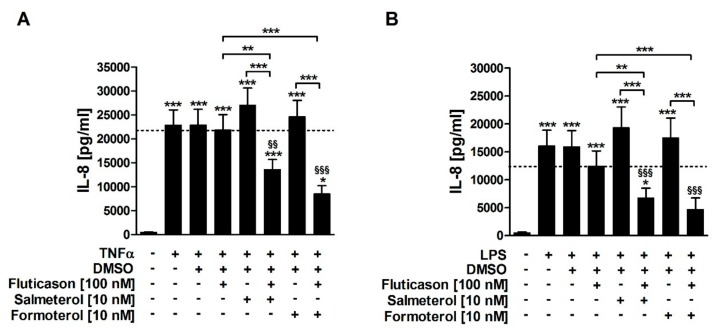
Both formoterol and salmeterol reverse the corticosteroid insensitivity of IL-8 in human airway smooth muscle cells (HASMCs) of subjects with chronic obstructive pulmonary disease (COPD). HASMCs (*n* = 5) of COPD subjects were treated with fluticason propionate (100 nM), formoterol (10 nM), salmeterol (10 nM) or with DMSO (0.45%; solvent control) two hours prior to stimulation with TNFα (20 ng/mL, (**A**)) or LPS (1 µg/mL, (**B**)). After 72 h of incubation, IL-8 was measured in culture supernatants by ELISA. Data are presented as mean ± SEM. One-way repeated-measures ANOVA with 95% CI: (**A**,**B**) *p* < 0.0001; post-hoc Bonferroni analyses: * *p* < 0.05; ** *p* < 0.01; *** *p* < 0.001 vs. unstimulated control or as indicated; ^§§^
*p* < 0.01; ^§§§^
*p* < 0.001 vs. TNFα + DMSO (A) or LPS + DMSO (B).

**Figure 2 jcm-08-02058-f002:**
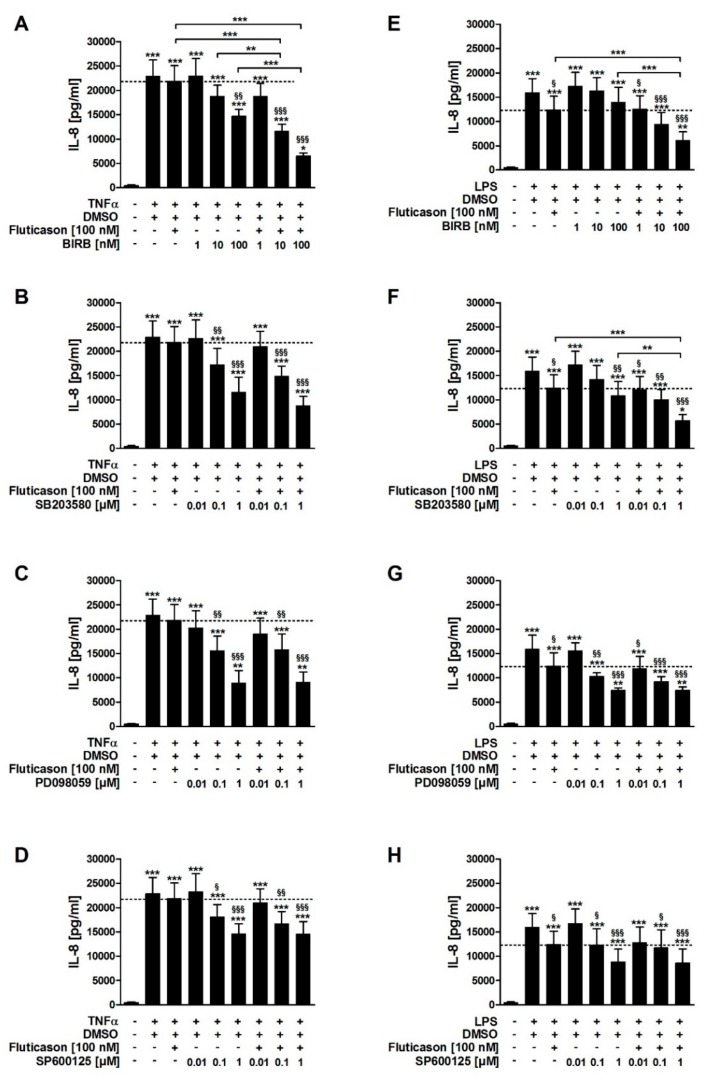
Effects of mitogen-activated-protein-kinase (MAPK) blockers on the corticosteroid insensitivity of IL-8 in HASMCs of COPD subjects. HASMCs (*n* = 5) of COPD subjects were treated with fluticason propionate (100 nM; (**A**–**H**)), BIRB796 (p38MAPKα, -γ, -δ inhibitor; (A,E)), SB203580 (p38MAPKα and -β inhibitor; (B,F)), PD098059 (inhibitor of ERK activation; (C,G)), SP600125 (JNK inhibitor; (D,H)) or with DMSO (0.45%, solvent control) two hours prior to stimulation with TNFα (20 ng/mL; (A–D)) or LPS (1 µg/mL; (E–H)). After 72 h of incubation, IL-8 was measured in culture supernatants by ELISA. Data are presented as mean ± SEM. One-way repeated-measures ANOVA with 95% CI: (A–H), *p* < 0.0001; post-hoc Bonferroni analyses: * *p* < 0.05; ** *p* < 0.01; *** *p* < 0.001 vs. unstimulated control or as indicated; ^§^
*p* < 0.05; ^§§^
*p* < 0.01; ^§§§^
*p* < 0.001 vs. TNFα + DMSO (A–D) or LPS + DMSO (E–H).

**Figure 3 jcm-08-02058-f003:**
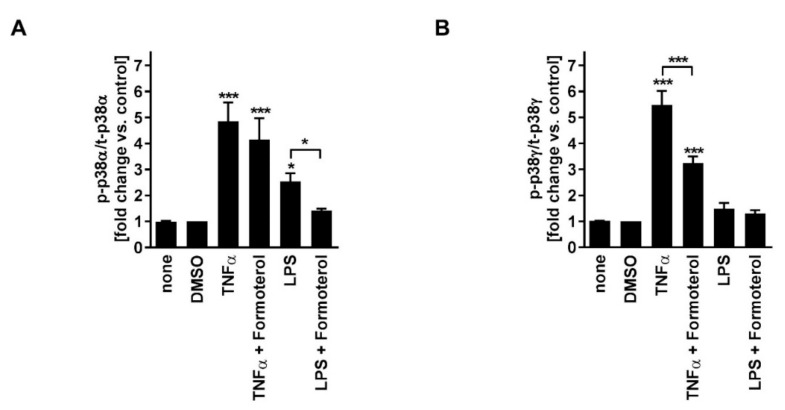
Formoterol reduces the activity of different p38MAPK isoforms depending on the inflammatory stimulus in HASMCs of COPD subjects. HASMCs (*n* = 4) of COPD subjects were treated with formoterol (10 nM) or with DMSO (solvent control) two hours prior to stimulation with TNFα (20 ng/mL) or LPS (1 µg/mL). After 15 min of incubation, levels of total and activated (T180 and Y182 phosphorylated) p38MAPKα (**A**) and of total and activated (T183 and Y185 phosphorylated) p38MAPKγ (**B**) were measured in cell lysates by ELISA. Data for phosphorylated proteins were normalized to the total levels and to the DMSO controls that were set to 1 (= no induction of activation). Data are presented as mean ± SEM. One-way repeated-measures ANOVA with 95% CI: (**A**,**B**) *p* < 0.0001; post-hoc Bonferroni analyses: * *p* < 0.05; *** *p* < 0.001 vs. DMSO control or as indicated.

**Table 1 jcm-08-02058-t001:** Effects of drugs and combinations on IL-8 expressed as % change. Data from Figure 1; Figure 2 were re-analyzed by calculating the drug effects on IL-8 as % change vs. stimulation with TNFα or LPS alone (each plus DMSO solvent control). Negative values show an inhibitory effect of the drug.

	% Change vs. TNFα (mean ± SEM)	% Change vs. LPS (mean ± SEM)
Fluticasone (100 nM)	−4.7 ± 1.4	−23.8 ± 3.9
Salmeterol (10 nM)	21.3 ± 12.9	22.1 ± 7.1
Formoterol (10 nM)	8.8 ± 6.6	10.2 ± 10.6
Fluticasone (100 nM) + Salmeterol (10 nM)	−38.0 ± 7.9	−59.5 ± 4.7
Fluticasone (100 nM) + Formoterol (10 nM)	−59.2 ± 8.4	−74.7 ± 5.9
BIRB (1 nM)	−0.4 ± 2.3	9.4 ± 3.5
BIRB (10 nM)	−16.3 ± 2.7	3.1 ± 4.4
BIRB (100 nM)	−33.7 ± 3.9	−13.7 ± 11.0
Fluticasone (100 nM) + BIRB (1 nM)	−15.5 ± 7.0	−23.2 ± 4.0
Fluticasone (100 nM) + BIRB (10 nM)	−46.9 ± 6.5	−43.8 ± 5.6
Fluticasone (100 nM) + BIRB (100 nM)	−69.5 ± 4.2	−64.1 ± 5.2
SB203580 (10 nM)	−2.9 ± 3.2	9.5 ± 3.2
SB203580 (100 nM)	−26.3 ± 5.6	−11.9 ± 4.0
SB203580 (1000 nM)	−51.9 ± 7.2	−35.1 ± 9.6
Fluticasone (100 nM) + SB203580 (10 nM)	−9.1 ± 1.3	−25.6 ± 5.1
Fluticasone (100 nM) + SB203580 (100 nM)	−34.8 ± 1.1	−37.2 ± 3.0
Fluticasone (100 nM) + SB203580 (1000 nM)	−62.5 ± 4.0	−63.5 ± 3.8
PD098059 (10 nM)	−11.4 ± 7.1	2.5 ± 8.5
PD098059 (100 nM)	−31.5 ± 8.9	−29.3 ± 9.1
PD098059 (1000 nM)	−58.8 ± 11.9	−48.1 ± 7.5
Fluticasone (100 nM) + PD098059 (10 nM)	−16.8 ± 6.1	−26.8 ± 3.4
Fluticasone (100 nM) + PD098059 (100 nM)	−31.4 ± 8.2	−38.7 ± 7.0
Fluticasone (100 nM) + PD098059 (1000 nM)	−58.6 ± 10.4	−49.6 ± 5.6
SP600125 (10 nM)	0.4 ± 3.7	5.1 ± 1.1
SP600125 (100 nM)	−19.8 ± 5.1	−26.2 ± 6.3
SP600125 (1000 nM)	−36.2 ± 1.8	−46.8 ± 8.1
Fluticasone (100 nM) + SP600125 (10 nM)	−7.9 ± 2.3	−22.9 ± 4.9
Fluticasone (100 nM) + SP600125 (100 nM)	−26.1 ± 6.4	−31.2 ± 8.2
Fluticasone (100 nM) + SP600125 (1000 nM)	−36.3 ± 6.0	−49.5 ± 10.4

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
