# Peer review of "LABAs and p38MAPK Inhibitors Reverse the Corticosteroid-Insensitivity of IL-8 in Airway Smooth Muscle Cells of COPD"

_jcm, 2019, doi:10.3390/jcm8122058_

Round 1
Reviewer 1 Report
The report by Knobloch et.al is well presented and well written. Whilst I believe the study has merit, I have some comments which require attention.
1. For the benefit of the reader who is not familiar with COPD remodelling processes, it would be useful if the authors briefly elaborated on what they are. For example lines 81 - 83:
'Human airway smooth muscle cells have been identified as an important source of COPD-related cytokines that link corticosteroid-resistant non-type 2 airway inflammation to remodeling processes'
What are the cytokines and what are the remodeling processes?
2. It would be useful to the reader if the authors provided some more information on their study population. For instance, what are the spirometry values (FEV1% and FEV1/FVC ratio) and what proportion are current / ex-smokers?
3. In the results section it would be beneficial to the reader to understand the magnitude of effects (or lack of). To help with this, insert the percentage changes into the results text. For example, fluticasone + LABA caused a XX% reduction in IL8.
4. The authors demonstrate a significant reduction in LPS stimulated IL-8 release by fluticasone (figure 2 graphs E, F, G, H). Is this the same data from figure 1B? If so where is the symbol to show a significant difference? In line with the previous comment, what is the percentage change?
I would argue that this is not corticosteroid resistant as the authors infer. This is merely insensitive. Other studies using lung macrophages (from both COPD and controls) have shown similar findings. Relative to the inhibition of other cytokines (TNFa / IL-6) IL-8 inhibition is insensitive (in a LPS stimulated model).
5. Do the authors have data on other cytokines?
Author Response
The report by Knobloch et.al is well presented and well written. Whilst I believe the study has merit, I have some comments which require attention.
For the benefit of the reader who is not familiar with COPD remodelling processes, it would be useful if the authors briefly elaborated on what they are. For example lines 81 - 83:'Human airway smooth muscle cells have been identified as an important source of COPD-related cytokines that link corticosteroid-resistant non-type 2 airway inflammation to remodeling processes'
What are the cytokines and what are the remodeling processes?
Answer: We agree and have added information about remodeling processes in general in the first paragraph and have added some details about the cytokines in the suggested paragraph.
It would be useful to the reader if the authors provided some more information on their study population. For instance, what are the spirometry values (FEV1% and FEV1/FVC ratio) and what proportion are current / ex-smokers?Answer: Thank you. We absolutely agree and have added spirometry data, age, gender, current/ex-smoker status and pack years at the beginning of the methods section.
In the results section it would be beneficial to the reader to understand the magnitude of effects (or lack of). To help with this, insert the percentage changes into the results text. For example, fluticasone + LABA caused a XX% reduction in IL8.Answer: We absolutely agree. However, we think that too many numbers/data in the text might make the reading more complicated. Therefore, we added the % change data in a table.
The authors demonstrate a significant reduction in LPS stimulated IL-8 release by fluticasone (figure 2 graphs E, F, G, H). Is this the same data from figure 1B? If so where is the symbol to show a significant difference? In line with the previous comment, what is the percentage change?Answer: Good point. We re-checked the analyses but found them to be correct. I have discussed the issue with the statistician. Yes, 1B and 2E-H are the same data regarding LPS controls and LPS + Fluticasone, and significant differences were in each case calculated with One-Way ANOVA plus Bonferroni post hoc test in Graph Pad Prism5. However, the data panels besides LPS controls and LPS + Fluticasone that were additionally analyzed with the ANOVA test are different in 1B (LPS +Salmeterol, LPS + Formoterol) vs. 2E, F, G, and H (LPS + MAPK Inhibitors in different concentrations and combinations), which influences the results of the post hoc tests. In this case, the same fluticasone effect was statistically minimal above p=0.05 in 1B and minimal below p=0.05 in 2E-H according to the Bonferroni post hoc tests. Summarized, same data, same test but according to the data that are additionally included in the ANOVA analyses the outcome can be slightly different regarding statistically significant differences around the threshold of p=0.05.
% Change data have been added in a table (see above)-
I would argue that this is not corticosteroid resistant as the authors infer. This is merely insensitive. Other studies using lung macrophages (from both COPD and controls) have shown similar findings. Relative to the inhibition of other cytokines (TNFa / IL-6) IL-8 inhibition is insensitive (in a LPS stimulated model).
Answer: We agree. We have changed it in this way in the whole text: we now use "insensitive" in the context of our cell culture models and "resistant" in the clinical/patient context.
Do the authors have data on other cytokines?The study has focused on IL-8 because it is central in corticosteroid-resistant non type 2 airway inflammation in COPD and because it is steroid resistant/insensitive in the HASMC culture model. Therefore, the samples (cultured HASMCs per well and supernatant) where rather small. We started to measure GM-CSF but found it to be steroid-sensitive in our models and stopped it because this did not match the question of the study. Meanwhile, the steroid-sensitivity of GM-CSF in our models has already been published in reference 7 so that adding GM-CSF data here would not have any significant value. So, we unfortunately don´t have data on other cytokines we could add. However, because of the important role of IL-8 we believe that the data do nonetheless support our conclusions.
Reviewer 2 Report
This short report assessed the effects of steroids and long acting beta agonists (LABA) on IL-8 expression in airway smooth muscle cells isolated from COPD lungs. IL-8 is known to be involved in neutrophilic-mediated inflammation in COPD and asthma. IL-8 expression was increased by TNF or LPS which was found to be resistant to FP treatment. However, the combination of FP and a LABA (formoterol or salmeterol) reduced TNF- and LPS-induced IL-8 expression. Additionally, the ability of MAPK (p38, ERK, and JNK) inhibitors with or without steroids to inhibit IL-8 was tested. While most of the MAPK inhibitors had moderate effects on IL-8, the combined effects of the p38 inhibitor, BIRB, and fluticasone were found to have the greatest effect on both TNF- and LPS-induced IL-8 expression. These data support clinical observations that using combined steroid and LABA therapy is a more effective treat for COPD and asthma. There is also evidence that p38 inhibition that specifically targets γ and δ isoforms may be a useful strategy. The relevance of MKP-1 to these data should be explored given the importance of MKP-1 mediated de-phosphoryation of p38 in the anti-inflammatory activity of steroids.
Major Comments
MKP1 has an established role in mediating the anti-inflammatory effects of steroids. GR increases MKP-1 expression leading to p38 de-phosphorylation. Accordingly, MKP1 expression and activity would be relevant to the fluticasone and p38 inhibitor interactions. Its unclear whether, p38 inhibition boosts steroid sensitivity or just inhibits IL-8 expression. Perhaps the combined effects of steroids, p38 inhibition and/or LABAs involve enhanced MKP-1 expression and activity. Could MKP-1 activity be deficient in COPD ASM? If so, this would support steroid resistance. In Fig 3, why was effects of only formoterol on p38 activity tested and not fluticasone and the p38 inhibitors? How does formoterol inhibit p38 activity? Is IL-8 expression in healthy or non-COPD ASM also steroid resistant?Author Response
This short report assessed the effects of steroids and long acting beta agonists (LABA) on IL-8 expression in airway smooth muscle cells isolated from COPD lungs. IL-8 is known to be involved in neutrophilic-mediated inflammation in COPD and asthma. IL-8 expression was increased by TNF or LPS which was found to be resistant to FP treatment. However, the combination of FP and a LABA (formoterol or salmeterol) reduced TNF- and LPS-induced IL-8 expression. Additionally, the ability of MAPK (p38, ERK, and JNK) inhibitors with or without steroids to inhibit IL-8 was tested. While most of the MAPK inhibitors had moderate effects on IL-8, the combined effects of the p38 inhibitor, BIRB, and fluticasone were found to have the greatest effect on both TNF- and LPS-induced IL-8 expression. These data support clinical observations that using combined steroid and LABA therapy is a more effective treat for COPD and asthma. There is also evidence that p38 inhibition that specifically targets γ and δ isoforms may be a useful strategy. The relevance of MKP-1 to these data should be explored given the importance of MKP-1 mediated de-phosphoryation of p38 in the anti-inflammatory activity of steroids.
Major Comments
MKP1 has an established role in mediating the anti-inflammatory effects of steroids. GR increases MKP-1 expression leading to p38 de-phosphorylation. Accordingly, MKP1 expression and activity would be relevant to the fluticasone and p38 inhibitor interactions.
Answer: We absolutely agree and this needs to be addressed in future studies. For this short report we think that investigating the underlying mechanisms are beyond the scope of the study. We just want to show the effects of the drug combinations in pre-clinical models 1) to explain effects of drug combinations in clinical trials (superiority of LABA/ICS vs. monotherapies) 2) to suggest in which direction clinical research activity might go (test if additional ICS use might be also beneficial in phenotypes without type 2 inflammation, and the future potential of isoform-specific MAPK inhibitors). For this, detailed underlying mechanisms would be interesting but - to our opinion - are not required at this point. And unfortunately we do not have samples left to check for MKP-1.
Its unclear whether, p38 inhibition boosts steroid sensitivity or just inhibits IL-8 expression.
Answer: Since BIRB is more effective in combination with fluticasone than alone (and is also superior to fluticasone) this additional effect should be beyond simple IL-8 blockade through BIRB in the TNF model (same for BIRB and SB in the LPS model). Support for the interpretation that formoterol reverses steroid insensitivity comes from the observation that formoterol blocks the activity of specific p38MAPK isoforms that have been found to be involved in steroid-resistance in PBMCS in severe asthma. We discuss these points in the 3rd and 4th paragraph of the discussion. We further added the sentence " These ex vivo data from severe asthma support our conclusions that the blockade of p38MAPK activity increases the sensitivity of IL-8 to fluticasone" to make it clearer that the literature supports our interpretation of the results.
Perhaps the combined effects of steroids, p38 inhibition and/or LABAs involve enhanced MKP-1 expression and activity. Could MKP-1 activity be deficient in COPD ASM? If so, this would support steroid resistance.
Answer: We absolutely agree. Unfortunately we do not have samples left from the same patients to test for MKP-1 expression. However, we added a passage regarding the possible role of MKP-1 in the paragraph 3 in the discussion and we added also reference 18 that shows the effects of LABAs on MKP-1 in HASMCs.
In Fig 3, why was effects of only formoterol on p38 activity tested and not fluticasone and the p38 inhibitors? How does formoterol inhibit p38 activity?
Answer: The point of Fig. 3 is to provide a first mechanistic explanation how formoterol might enhance fluticasone effects on IL-8. It is known from refs 16 and 17 that LABAs might block specific p38 isoforms (delta, alpha) to improve steroid effects in systemic cells. Therefore, we tested whether formoterol might do the same in this culture model of lung cells. And since the mechanisms seem to be similar, we concluded that indeed isoform-specific p38 inhibitors and formoterol improve fluticasone effects. p38 inhibitor effects on p38 activity were not tested in this study because their effects have been described multiple times. Fluticasone effects on p38 activity would be interesting but would not help to answer the questions of the study. Formoterol might block p38MAPK isoforms via MKP-1 upregulation - this possible explanation has now been added to the discussion (see above).
Is IL-8 expression in healthy or non-COPD ASM also steroid resistant?
Answers: Thank you very much for this point. We forgot to mention this for the LPS model in the introduction. In ref. 7 we have shown that IL-8 is sensitive to fluticasone in HASMCS of non-COPD subjects in both models (LPS and TNF). For this reason, the 2 HASMC models have great COPD clinical/therapeutical relevance, because they mimic the COPD-specific IL-8 insensitivity to steroids in COPD. We have added the information in the introduction (paragraph 7)
Reviewer 3 Report
Main comments:
Although the concept according to which the p38MAPK inhibitors and LABAs reverse steroid insensitivity is well know, the manuscript result very interesting in that it highlights IL-8 role and different p38MAPK isoforms role in different stage of COPD.
The study population is composed by patients with both COPD and Lung Cancer, while all the manuscript is focused only on COPD. Could cancer, in some way, play a role or influence the pathway considered? Are there differences compared to the patients with only COPD? HASMCs are pretreated with fluticason, formoterol or salmeterol two hours prior to TNFα or LPS stimulation. Could the authors explain the reason? Could the pretreatment not represent a culture model of stable or exacerbated COPD? It’s like to treat the patient before he has the pathology. Why 72 hours incubation time is chosen to reveal IL-8? Because IL-8 is NF-kB dependent, which is the role of NF-kB in the experimental conditions, after Fluticason or LABAs addition? Formoterol reduces the activity of different p38MAPK isoforms. Is it known a molecular mechanism? Are there references in this regard? Which is the role of β2 receptors?
Minor comments:
Lines 5-6: check the character
For greater clarity and ease of reading, I advise the authors to divide the paragraph of the results into subparagraphs.
Figure legends must be clearer; please, divide the descriptions (for example for Figure 1A and 1B)
Could the authors specify the DMSO percentage used?
Author Response
Main comments:
Although the concept according to which the p38MAPK inhibitors and LABAs reverse steroid insensitivity is well know, the manuscript result very interesting in that it highlights IL-8 role and different p38MAPK isoforms role in different stage of COPD.
The study population is composed by patients with both COPD and Lung Cancer, while all the manuscript is focused only on COPD. Could cancer, in some way, play a role or influence the pathway considered? Are there differences compared to the patients with only COPD?
Answer: Good point but unfortunately we cannot give a definitive answer on this. HASMCs were isolated from lung resections for which lung cancer diagnosis is a prerequisite. This is an established method to obtain HASMCs but nonetheless you are right. We just want to remind that the cells were isolated from "healthy" tissue and the analyses have been done in culture passages 2-5 suggesting a strong reduction of the influence of surrounding cancer tissue. Moreover, IL-8 in HASMCs of lung cancer patients without COPD is steroid-sensitive (reference 7), demonstrating that the steroid-insensitivity of IL-8 in HASMCs results from COPD. However, we cannot fully exclude an influence of lung cancer. We stated this in the text.
HASMCs are pretreated with fluticason, formoterol or salmeterol two hours prior to TNFα or LPS stimulation. Could the authors explain the reason? Could the pretreatment not represent a culture model of stable or exacerbated COPD? It’s like to treat the patient before he has the pathology.
Answer: Good point that needs to be discussed. 1. We used pre-treatments to obtain maximal effects, which is important to analyze for differences between several drug combinations. 2. the cells are from COPD subjects meaning that they have the molecular pathologies associated with COPD before any treatment in culture; for example: we have shown COPD-specific steroid-insensitivity in pre-treatments (here and refer to ref. 7). This demonstrates that there is already a COPD molecular pathology in the cultured cells before any treatment in culture. Furthermore, I would like to argue that TNFalpha and LPS addition to the culture medium might just be a "re-stimulation" of the cells, because the cells should have already been exposed to these factors in vivo (before lung resection and cell isolation) many times before (because the permanent or repeated exposition of HASMCs to increased TNF and to LPS are part of the COPD pathophysiology). Following these thoughts, we think that the addition of drugs before TNF and LPS in culture cannot be interpreted as "treating a patient before he has the pathology" but has clinical relevance. I think for cultivated HASMCs obtained from COPD subjects, the issue of a pre-treatment with drugs is completely different to the discussion on mouse models where this issue is well addressed. There you have healthy mice that get exposed to stimulants that induce the disease. In this case pre-treatments are critical because this would indeed mean that you would treat healthy subjects before they get the disease. However, here the cells are from COPD subjects that have already COPD-associated molecular pathologies.
Why 72 hours incubation time is chosen to reveal IL-8?
Answer: as stated in the methods section, 72 hours of incubation is an established condition for HASMCs (references 7 and 8). After 72 hours there is maximal cytokine release in response to the used stimulants.
Because IL-8 is NF-kB dependent, which is the role of NF-kB in the experimental conditions, after Fluticason or LABAs addition?
Answer: This is an interesting question that, however, is beyond the scope of our study and needs to be addressed in future studies.
Formoterol reduces the activity of different p38MAPK isoforms. Is it known a molecular mechanism? Are there references in this regard? Which is the role of β2 receptors?
Answer: As suggested by reviewer 2, the possible mechanism might be via the up-regulation of MKP-1 by LABAs. This has been mechanistically investigated for HASMCs in an excellent in vitro study which is now cited as reference 18.
Minor comments:
Lines 5-6: check the character
Answer: Sorry, we do not understand the point. Line 5-6 on which page? We have checked all pages and did not find anything.
For greater clarity and ease of reading, I advise the authors to divide the paragraph of the results into subparagraphs.
Answer: We have already three subparagraphs in the results section that each includes the data for one train of thought. Subparagraph 1 describes the effects of the beta-2 agonists alone and in combination with fluticasone on IL-8 sowing that LABAs can improve steroid effects. Subparagraph 2 describes the effects of fluticasone and MAPK inhibitors on IL-8 and the effects of formoterol on MAPK activity in the TNFa model showing that p38gamma blockade improves steroid effects there. Subparagraph 3 describes the same for the LPS model showing that p38alpha blockade improves steroid effects in this model. We do not know how to further divide these subparagraphs without loosing the context. We would be very happy to get any suggestions.
Figure legends must be clearer; please, divide the descriptions (for example for Figure 1A and 1B)
Answer: Experiments in 1A and 1B are identical with the exception of the stimulants (TNF in A, LPS in B). If we would write separate texts for both panels we would have 2 times nearly identical text - with the exception of the stimulant. We do not think that this might be helpful for the reader. So, we have inserted the panel code (A or B) after the stimulant. The same we have done for the drugs in Figure 2 since A-H differs regarding type of MAP inhibitors and stimulant and for Fig. 3. Does this help? If not, we are very sorry but then we did not get the point. Could you please tell us details about what is unclear in the legend of Fig. 1?
Could the authors specify the DMSO percentage used?
Answer: Thank you. We forgot to mention this. It was 0.45 %. We added this information to the figure legends.
Round 2
Reviewer 2 Report
I have no further comments.
Reviewer 3 Report
No further comments